# Neuromuscular Stability of Dental Occlusion in Patients Treated with Aligners and Fixed Orthodontic Appliance: A Preliminary Electromyographical Longitudinal Case-Control Study

**DOI:** 10.3390/diagnostics12092131

**Published:** 2022-09-01

**Authors:** Claudia Paola Bruna Dellavia, Giacomo Begnoni, Cristiana Zerosi, Guia Guenza, Natalie Khomchyna, Riccardo Rosati, Federica Musto, Gaia Pellegrini

**Affiliations:** 1Department of Biomedical Surgical and Dental Sciences, University of Milan, Via Luigi Mangiagalli 31, 20133 Milan, Italy; 2Independent Researcher, Via Matteo Bandello, 6, 20123 Milan, Italy

**Keywords:** ssEMG, Invisalign, aligner, fixed appliance, orthodontics, muscular function

## Abstract

The aim of the present study was to evaluate if, after treatment with aligners (ALIGN) and fixed orthodontic appliance (FOA), alterations of the neuromuscular activity may occur and if differences in these changes can be detected between the two treatments. Sixteen healthy patients (7 FOA, 9 ALIGN) with class I or class II molar relation were recruited. Standardized surface electromyography (ssEMG) was used to evaluate the activity of the masticatory muscles (masseters-MM and temporalis-TM) before the beginning of the orthodontic treatment (T1), at the end (T2), and 3 months (T3) after the end of the treatment. Intragroup (within timepoints) and intergroup differences were statistically analyzed. At T1, the mean values of each ssEMG index were within the normal range in both groups. At T2, the FOA group showed larger differential recruitment of the MM than TA muscles with a value slightly over the normal range. All the indexes were normalized at T3, and no differences emerged between groups. In the FOA group, the index of MM symmetrical contraction increased significantly at T3 compared to T1 and T2. In the ALIGN group, no significant changes were observed between each timepoint. In FOA subjects, a slight alteration of the muscular activity appeared immediately after bracket removal and this alteration normalized after 3 months of rescue. In subjects treated with aligners, no significant alteration of the muscular activity was assessed.

## 1. Introduction

In the last 20 years, a request to expand the choice of invisible orthodontic devices in response to the patient’s growing aesthetic needs has led to a significant increase in the supply of invisible aligners. 

The range of cases in which aligners can be applied has been the subject of much discussion. They were born as an orthodontic alternative for adult patients with Angle class I malocclusion and mild to moderate crowding but, to date, even more complex cases including extractions [1,2], class II subdivision [3], openbite [4], surgical class III [5], and interceptive treatment in children [6] have been presented.

Regarding the occlusion, the traditional fixed orthodontic appliance (FOA) contemplates the maintenance of the direct intercuspation between maxillary and mandibular teeth. The teeth are bound to the metal arch and splinted together. To the contrary, the aligners are made up of masks with an occlusal component that interposes between maxillary and mandibular teeth and that excludes their intercuspation except for limited periods of time during the day when the aligners are removed for meals and oral hygiene procedures. However, in one out of three cases, molars struggled with buccal crown tip likely because of poor aligner grip around the shorter terminal crown and the decreased forces on the terminal tooth within the aligner which results in vestibular flaring of the dental elements avoiding mutual interaction between the antagonistic teeth, thus precluding the possibility of occlusal adjustments during therapy [7].

According to a recent meta-analysis, the setting of occlusal contacts obtained by using aligners is worse compared to FOA [8]. Considering these aspects, at the end of the treatment, the two orthodontic approaches may lead to important differences in the occlusal stability and in the function of the related neuromuscular system. However, this aspect has not been solved yet.

Non-standardized surface EMG investigations have been conducted during or after treatment with functional devices [9,10] and FOA [11,12,13] and assessed that tooth movement changes the muscular activity. The use of a non-standardized method for the obtainment of data comparable in clinical and research activity is still controversial [14]. Standardized surface electromyography (ssEMG) is an objective and useful instrumental approach in the analysis of dental occlusion by recording the signals of activity of masticatory muscles [15,16,17]. This standardized methodology is based on the use of cotton rolls to exclude the influence of occlusal contacts on the periodontal receptorial system and then on the muscular activity. Studies reported that even small intercuspal interference, such as an aluminum of 0.25 mm of height, could give rise to asymmetric contractile activity in the mandibular elevator muscles studied by means of ssEMG, as well as potentially displacing the mandible in a lateral direction [18]. The clinical application of ssEMG has already been observed in different populations by monitoring the effects of occlusal splint [19], myofunctional therapy [20], and prosthetic rehabilitation [21,22] on occlusal stability. However, ssEMG data about occlusal stability and equilibrium, as well as muscular activity achieved at the end of both the orthodontic therapies are limited. Regarding the study of the effects of FOA on the masticatory muscles’ activity, the ssEMG has proved to be useful in intercepting the risk of dental relapse as evidenced by altered electromyographic values in a previous case series [23]. However, a standardized sEMG protocol has never been adopted for the evaluation of the efficiency of aligners and FOA at producing a stable and comfortable dental occlusion at the end of the treatment.

The aim of the present study was to assess, by means of ssEMG, if, after treatment with aligners and FOA, immediate or prolonged alterations of the neuromuscular masticatory activity may occur and if differences in these neuromuscular changes can be detected between the two treatments.

## 2. Material and Methods

The present work is a preliminary observational longitudinal case-control study of two groups of patients presenting similar occlusal characteristics that underwent two different orthodontic therapies and whose muscular response was analyzed with a standardized surface electromyographic analysis.

### 2.1. Patient Recruitment

Sixteen patients were selected consecutively at the private practice of C.Z. during a period of 3 years from September 2016 to December 2019 by respecting the inclusion and exclusion criteria. The patients selected for this study were followed for the treatment until July 2020. All patients were followed at least until the period of 3 months after the end of the treatment. All clinical data were collected by two operators (G.G. and K.N.). After a detailed explanation of the experimental protocol, all patients (or parents/legal guardians when <18 years) were asked to participate in the study and to fill and sign an informed consent to all the clinical and EMG procedures. The study protocol was approved by the ethics committee of the University of Milan (DG-EMG-2016).

Seven patients were treated with FOA (FOA group), and nine patients with Invisalign^®^ (ALIGN group).

All the clinical diagnoses, the definition of the treatment plan, and the therapy were performed to all patients by the same clinician, C.Z., orthodontist specialist and Invisalign provider. Each treatment plan was always discussed together with the operator. Since there were no differences in the costs between the two therapies and the clinical dental and skeletal characteristics of all patients were similar, the decision for a FOA or ALIGN treatment was made by the patients (or the parents in case of underage patients) according to personal preferences. 

For the FOA group, traditional vestibular multi-bracket systems (022 braces–MBT prescription) were placed. The arches sequences used were: 014, 018 Nitinol archwire for the alignment phase; 014 × 025, 019 × 025 Nitinol, and 019 × 025 stainless steel archwires for the leveling phase; and 019 × 025 TMA for the finishing phase. Neither posterior nor anterior occlusal build-ups were placed. Intermaxillary elastics and bends were provided in the last phase of treatment, if necessary. 

For the ALIGN group, a comprehensive series of aligners was requested, and for all patients at least one series of additional aligners was requested to refine the finishing phase of treatment. The type, number, and position of the attachments depended on the personal choice of the operator.

All patients underwent clinical and ssEMG examination to verify correspondence with the inclusion criteria before being enrolled in the study. Inclusion and exclusion criteria were created considering Joffe’s (2003) suggestions [24]. 

Inclusion criteria were no skeletal transverse deficiency of the arches, craniomandibular divergence (S.N/Go.Gn) between 27° and 37°, between mild to moderate crowding (1–5 mm), mild to moderate spacing (1–5 mm), absence of systemic diseases, absence of stomatognathic apparatus or neck muscular disorders, presence of at least 28 permanent teeth, absence of periodontal disease, absence of craniofacial traumas or temporomandibular joint disorders, absence of peripheral nerve and muscle diseases, an acceptable level of compliance, and ssEMG values within the normal range [15,25]. 

Exclusion criteria were orthodontic therapies in progress, oral surgeries in the previous 3 months, medications that interfere with the musculoskeletal system, centric occlusion and centric relation discrepancies, any signs of symptoms regarding the temporomandibular disorders, skeletal anteroposterior discrepancies of more than 2 mm (as measured by discrepancies in cuspid relationships), severely rotated teeth (>20°), severely tipped teeth (>45°), and arches with multiple missing teeth. All the patients were followed since the beginning of the therapy by an oral hygienist in order to maintain the periodontal health under control.

Extraoral and intraoral documentation were recorded for each patient before and after completion of orthodontic therapies with both aligners and FOA. At the end of the orthodontic therapy, a retention wire in the lower jaw and a Hawley plate in the upper jaw were applied to all the participants of the study.

### 2.2. Electromyographic Analysis

The electromyographic protocol, electrode types, measurements, and procedures have been explained in a previous article [15].

#### 2.2.1. Electrode Type and Positioning

Anterior temporalis (TA) and masseter (MM) muscles were bilaterally examined. Disposable pregelled silver/silver chloride bipolar surface electrodes (rectangular shape, 21 × 41 mm, 20 mm interelectrode distance) (F3010, Fiab, Firenze, Italy) were placed in correspondence of the projection of the superficial belly of the masseter muscle and the anterior bundle of the temporalis muscle (Figure 1) as follows:MM: the electrodes were fixed parallel to the exocanthion–gonion line and with the upper pole of the electrode under the tragus–labial commissural line. The operator, standing in front of the seated patient, checked the muscular belly with palpation while the patient clenched his/her teeth.TA: the muscular belly was palpated during tooth clenching and the electrodes were fixed vertically along the anterior margin of the muscle (corresponding to the frontoparietal suture).

The skin was carefully cleaned with alcohol prior to electrode placement to reduce skin impedance, and there was a 5 min waiting time to allow the conductive paste to adequately moisten the skin prior to proceeding with the recordings.

#### 2.2.2. Instrumentation

A computerized electromyograph was used to record the EMG activity (EasyMYo, 3 Technology S.r.l., Udine, Italy) in order to analyze the activation pattern of the MM and TA muscles. A differential amplifier with a high common-mode rejection ratio (CMRR = 100 dB, in the range of 0–60 Hz, input impedance 100 GΩ) was used to amplify the recorded analog sEMG signal (gain 100, bandwidth 0–1000 Hz, peak-to-peak input range from 0 to 3600 mVpp). It was then digitized (24-bit resolution, 4000 Hz A/D sampling frequency) and digitally filtered (Butterworth type, high-pass filter set at 30 Hz, low-pass filter set at 400 Hz, band-stop for common 50–60 Hz noise). The signals were averaged over 25 ms intervals, with muscle activity assessed as the root mean square (RMS) of the amplitude (µV). ssEMG signals were recorded for further analysis. Before the acquisition session, the patients were properly trained to perform true maximal voluntary contraction using live ssEMG signal visualization.

#### 2.2.3. Measurements

The following two steps were performed during each recording session (T1, T2, and T3):Masticatory muscles standardization procedure: two 10 mm thick cotton rolls were positioned on the mandibular second premolars/first molars of each patient, and a 5 s maximum voluntary clenching (MVC) was recorded to standardize TA and MM sEMG signal. The mean ssEMG potential obtained in the first acquisition was set at 100%, and all further ssEMG potentials were expressed as a percentage of this value (µV/µV × 100) [15].Maximum voluntary teeth clenching (MVC): TA and MM ssEMG activity was recorded during a 5 s MVC test in intercuspal position; the patients were invited to clench as hard as possible and to maintain the same level of contraction during the entire test.

For each acquisition, the best performance 3 s intervals were isolated and analyzed. During the tests, patients were asked to perform at their best, to avoid head and neck movements, and, to maintain a relaxed facial expression to reduce cross-talks. Within patients, the test’s order between the standardization procedure and MVC test was randomized, and an adequate period of rest was allowed. All acquisitions were made by the same operator [G.B.].

Maximum voluntary clench is a tooth-determined position which is defined by the position of the mandible when the relationship of opposing occlusal surfaces provides for maximum planned contact and/or intercuspation [26].

#### 2.2.4. ssEMG Data Analysis

The ssEMG potentials recorded during the MVC test in intercuspal position were expressed as a percentage of the mean potential recorded during the standardization test (MVC on cotton rolls), unit: µV/µV × 100 [15]. 

All further calculations were based on the standardized potentials. A set of the following ssEMG indexes were then computed [16]:Percentage overlapping coefficient (POC %): it compares the ssEMG waves of paired (left and right, masseter and temporalis) muscles to evaluate muscle symmetry; it ranges between 0% and 100%, as determined by occlusion. When two paired muscles contract with perfect symmetry, a POC of 100% is obtained. There were 95% of subjects without muscular imbalances of dental origin who had POC values between 80 and 90% [15].Asymmetry index (ASIM %): it compares the influence of dental contacts on the total activity of the right MM and TA with respect to the left MM and TA; a negative value indicates a greater differential activity of the left antimere; a positive value indicates a greater differential activity of the right antimere. There was a total of 95% of subjects without muscular imbalances of dental origin having values of asymmetry between ± 10%.Activation index (ACTIVITY %): it compares the influence of dental contacts on the TA muscle activity vs. MM muscle activity; it is calculated as the percentage ratio of the difference between the mean masseter and temporalis anterior muscles’ standardized potentials and the sum of the same standardized potentials, to individuate the most prevalent pair of masticatory muscles. A negative value implies greater differential recruitment of TA muscles, while a positive value implies greater differential recruitment of the MM muscles. There was a total of 95% of subjects without muscular imbalances of dental origin having activation values between ± 10% [27].Torque coefficient (TORQUE %): it evaluates the potential lateral displacing component; this component could derive from an unbalanced contractile activity of contralateral masseter and temporalis muscles, for example, right MM and left TA. This index ranges between 100% (complete presence of the right temporalis anterior and left masseter muscles) and −100% (complete prevalence of the left temporalis anterior and right masseter muscles). There was a total of 95% of subjects without muscular imbalances of dental origin having torque values between ±10%.Total muscular activity index (IMPACT %): it quantifies the total muscular activity performed during MVC relative to the standardization clenching on cotton rolls; IMPACT is estimated by computing the mean (masseter and temporalis anterior) total muscle activities as the integrated areas of the ssEMG potentials over time. There was a total of 95% of subjects without muscular imbalances of dental origin having IMPACT values between 80 and 120% [25].

The information elaborated by data acquisition (DAQ) software for differential analysis of muscle activity was obtained by the standardized electromyographic signals’ registration (Figure 2). 

### 2.3. Study Design

Standardized superficial myoelectric signals of the right and left temporalis and masseter muscles were recorded during maximum voluntary teeth clenching (MVC). Three examination sessions have been held as follows:T1: before the onset of the orthodontic treatment;T2: at the end of the orthodontic treatment (just after the debonding or attachment removals and the placement of the retention wire in the lower jaw and just before the placement of the Hawley plate in the upper jaw);T3: after 3 months of retention.

### 2.4. Statistical Analysis

The values obtained from all timepoints (T1, T2, and T3) of both groups were analyzed for descriptive and inferential statistics. All ssEMG indexes underwent analysis with two-way ANOVA with repeated measures on one factor (F1: treatment, F2: timepoint) followed by post hoc *t*-Tests [28]. The significance was set at *p* < 0.05. 

### 2.5. Sample Size Calculation

By assuming to detect a significant statistical difference in the POC values between the 2 groups of 4.59% with a standard deviation of 5.24 as previously reported by Ferrario [18], a margin error of 22.8 was calculated. Considering a power of 80%, alpha 0.05, beta 0.02, a sample size of 30 was detected. Considering the nature of this study, preliminary data in a sample of 16 patients are herein presented.

## 3. Results

### 3.1. Clinical Data

The mean age, sex, orthodontic characteristics, and mean duration of the treatment in the two groups are summarized in Table 1. No statistical differences were observed for any of the cephalometric indexes at T1 between the two groups. A significant difference (*p* < 0.05) was observed in the mean ages between the two groups. Adult patients opted for ALIGN therapy (mean age 25.60 ± 13.17 years); when the choice had to be made by parents, they always preferred the traditional brackets therapy, and, therefore, the mean age of FOA was significantly lower (12.49 ± 1.07 years).

Before concluding the therapies, all the patients of both groups were evaluated by the operators (G.G. and N.K.): functional mandibular movements were evaluated clinically by asking the patients to protrude and to move laterally the lower jaw, neither deflections nor impediments were observed;occlusal stability was evaluated by asking the patients to clench on articulating paper (blu articulating paper 200µ Bausch, Nashua USA) and to bite hard; no shift or deflections were noticed, and symmetrical distribution of occlusal contacts was observed.

The time duration of the therapy was significantly different between the two groups (*p* < 0.05): 23.86 ± 7.69 months for the FOA group and 15.33 ± 7.78 months for the ALIGN group. At the end of the orthodontic therapy, a bilateral molar and canine Angle class I relation was achieved in all the patients participating in the study. Additionally, overjet and overbite were within normal range values at the end of the therapy.

### 3.2. Electromyographical (ssEMG) Assessment

Table 2 shows the descriptive and inferential statistics of the standardized indexes obtained before (T1), at the end (T2) of the treatment, and at 3 months (T3) of follow-up. 

ssEMG analysis at treatment onset (T1). Before starting the orthodontic therapy, the mean values of each parameter were within the normal range in both groups even if a few single data results deviated in some patients of both groups as reported below. 

The symmetry indexes POC and ASIM were within the normal range for all couples of muscles in all but one patient of the FOA group and two patients of the ALIGN group. The IMPACT of anterior temporalis and masseter muscles for both groups was within the normal range in all but one other patient of the FOA group. 

The barycenter (ACTIV) results centered in the antero-posterior direction with border line values only for two FOA patients. Regarding the TORQUE, one patient of the FOA group had values over the normal range.

At the statistical comparison between the two treatments, the ACTIV index results were only significantly higher in subjects treated with FOA, close to the high limit of the normal range. 

ssEMG analysis at the end of treatment (T2). At the end of the orthodontic therapy and immediately after the removal of brackets or of the last aligner, the mean values of each parameter resulted within the normal range in both groups, but the mean ACTIV in the FOA group (15.1%) slightly exceeded the higher limit of normal range. 

At the observation, a few single data results deviated in some patients of both groups as reported below. 

One patient of group FOA and two of group ALIGN showed a POC TA slightly under the normal range. The percentage overlapping coefficient for masseters in the FOA results were slightly under 80% in two patients and very low (69%) in one patient; in the ALIGN group, only one value resulted slightly under the normal range. The IMPACT resulted over the normal range in three FOA patients and under the reference value in one FOA patient. In the ALIGN group, all IMPACT values were within the normal range. Even the ASIM and ACTIV presented values over the normal range in one FOA subject. 

At the intragroup comparison of timepoints (T1 vs. T2), patients treated with FOA showed a slight reduction in POC MM (from 85.1% at T1 to 80.4% at T2), and an increase in IMPACT (from 92.8% at T1 to 113% at T2) and increase in ACTIV (from 9.5% at T1 to 15.1% at T2) compared to the initial status. To the contrary, when the last aligner was removed, patients presented an increase in POC MM (from 83.2% at T1 to 86.0% at T2) and decrease in IMPACT (from 102.2% at T1 to 95.9% at T2). Similar to FOA, the align group showed an increase in ACTIV values (from 3.9% at T1 to 6.8% T2). 

The other indexes (POC TA, POC mean, TORQUE) remained mostly stable between the two timepoints in both groups. 

At the inferential statistics, the FOA group IMPACT at T2 results were significantly higher than at T1 but still within the normal range. No differences emerged between the groups in any of the analyzed parameters.

ssEMG analysis 3 months after the end of treatment (T3). After 3 months of nocturnal mobile retention, the mean values of each parameter resulted within the normal range in both groups. At the observation, a few single data results deviated in some patients of both groups as reported below. 

The FOA group gained a normalization of the differential recruitment of temporalis and masseter muscles, while two patients of ALIGN showed, respectively, a slightly lower POC TA and POC MM. The IMPACT results were slightly high in one FOA and in two ALIGN. ASIM was slightly high in one patient of both the FOA and ALIGN groups, and high in one ALIGN. One ALIGN patient also showed high ACTIV value. TORQUE was overall in the normal range in all patients. 

At inferential statistics, no differences between the groups appeared for all parameters. 

The assessment of differences between timepoints (T1 vs. T2) revealed that the FOA group values of POC MM at T3 increased significantly compared to both T2 and T1 (*p* < 0.05). Values at T3 further increased compared to T2 that were in the lower part of the range of normality, outdoing even the already optimal T1 values.

## 4. Discussion

The present study investigated the effect of dental occlusion on the neuromuscular system in patients treated with orthodontic devices that differently affect the dental intercuspation. For this purpose, data were harvested by means of standardized surface EMG protocol before, immediately after, and 3 months after the completion of orthodontic therapy performed using FOA or aligners.

Changes of electromyographic data observed in each group during the different timepoints showed that at the end of the treatment, the two systems seem to differently affect the proprioception. In subjects treated with aligners, all ssEMG parameters remained stable during all the period of study. No significant alteration was observed in muscular activity at the aligner removal (T2) and after the 3 months of rescue (T3) compared to baseline. This data disagrees with the literature that observed worse occlusal contacts in patients treated using aligners compared to FOA [25]. The removal of the aligner for short periods of time may be sufficient for the maintenance of the memory of the proprioception.

Differently, in patients treated with FOA, mean value of the total muscular activity (IMPACT) collected immediately after removal of the fixed device was 113.5%. This data is in the normal range and close to the higher limit, but the results were significantly higher compared to that obtained before the beginning of the orthodontic therapy (92.8%). After the rescue period, this parameter decreased (100.4%) and settled at an intermediate level between T1 and T2.

Furthermore, the mean value of the antero-posterior barycenter (ACTIV) measured at T2 (15.1%) just exceeded the higher limit of the normal range [15] and then returned in the physiological range at T3. These data seem to indicate that when patients perform clenching immediately after FOA removal, a high total muscular activity is required to maintain the occlusion. This muscular recruitment mainly involved the masseters, thus, moving the barycenter of the force in the posterior area. Differently from patients treated with aligners that can remove the device for short periods during the entire orthodontic therapy, in the FOA group after bracket removal, subjects perceive a sudden tooth instability [26] that during clenching may induce the higher recruitment of muscles involved in mandible stabilization. This response may cause the temporary less efficient and imbalanced activity of their neuromuscular system as also indicated by the decrease in MM symmetry below the normal range at T2. These findings are in agreement with studies from other authors [12,29] that reported significant alterations in the electrical activity of the masseter muscle assessed during orthodontic fixed treatment: they assumed that these changes were due to discomfort or pain or to changes in the occlusal relationship between the maxillary and mandibular dentitions, thus providing new periodontal afferents that may negatively influence the neuromuscular equilibrium. Proffit reported the solving of this occlusal and neuromuscular instability a few months after bracket removal [26]. For this reason, in the present study after 3 months of nocturne passive retention, which granted a certain degree of occlusal accommodation, the FOA group gained ssEMG values within the physiological range of healthy subjects. These data confirm that physiological normalization of the neuromuscular system can be observed already at 3 months after the end of the treatment.

In this study, to evaluate changes in the neuromuscular activity, the ssEMG values taken at baseline were considered as reference for comparison with the following timepoints since all patients included in the study were without disorders at the stomatognathic apparatus or neck muscles. Although the present work is focused on a small sample of patients and a high interindividual variability can be found [30], it is interesting to observe that no altered values of muscular imbalance have been detected in both groups at T3. This could suggest that the effects of orthodontic treatments in patients with simple/mild malocclusion, when correctly planned and monitored, are associated to a maintenance of a neuromuscular stability [31,32]. Future studies with a larger sample size are needed to draw a more reliable and supported conclusion about neuromuscular responses during orthodontic treatment performed by means of aligners or traditional fixed appliance even on subjects with functional disorders.

According to Proffit, neuromuscular adaptation is crucial in obtaining occlusal stability over time [26]. Otherwise, relapse may be seen as a “physiologic” attempt to return to an acceptable neuromuscular equilibrium [33]. The masticatory system attempts to re-establish occlusal stability through compensatory mechanisms. In this regard, ssEMG has revealed itself to be a reliable instrument to investigate the neuromuscular balance provided by dental occlusion [15]. Evidence about the use of ssEMG in teeth alignment therapies, whether they are traditional or invisible, is still lacking. However, previous studies have suggested the possibility to investigate the muscular response of masticatory muscles in patients that underwent orthodontic treatment [34]. According to these previous works, surface electromyography can help clinicians in the monitoring of the occlusal assessment phase at the end of the orthodontic treatment when using a standardized protocol. Moreover, in the case series presented by Ferrario et al., altered electromyographical values appeared linked to the risk of dental relapse after the orthodontic therapy had finished [18].

Within the limitations of the present study is the distribution of the younger population mainly in the FOA group and of adults in the ALIGN group. However, at baseline, the younger patients of the FOA group and the older patients of the ALIGN group showed similar values of muscular activity. In fact, the electromyographical data are presented as a percentage obtained by comparing the MVC and the standardized activity of the masticatory muscles. In this way, the standardized data are not influenced by age [35].

In the present work, the shorter duration of therapy with aligners is mostly linked to the different maintenance in terms of oral hygiene showed by patients treated with FOA with consequent brackets detachment and delayed treatment duration. Moreover, the ALIGN patients showed an excellent compliance for the whole duration of the treatment. This can also be linked to the different age of the patients in the two groups: older patients treated with aligners showed higher compliance than younger patients treated with traditional brackets.

## 5. Conclusions

To conclude, both therapies seem to lead the patients toward a general good muscular balance inset.

Standardized surface electromyography allowed the assessment of the neuromuscular system in patients treated with fixed orthodontic appliance or with aligners. Data showed that in FOA subjects, a slight alteration of the muscular activity appeared immediately after bracket removal and that this alteration returned in the normal range after 3 months of rescue. In subjects treated with aligners, no significant alteration of the muscular activity was assessed. The ssEMG can help clinicians in the monitoring of the occlusal assessment phase at the end of the orthodontic treatment.

## Figures and Tables

**Figure 1 diagnostics-12-02131-f001:**
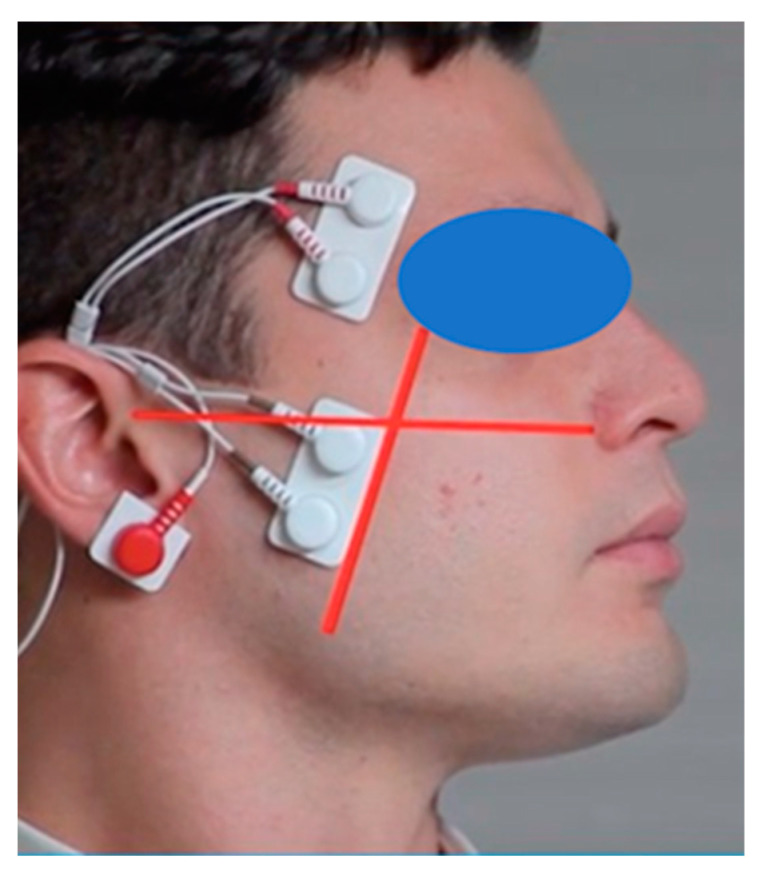
Masseter and temporalis anterior muscles’ electrode placement.

**Figure 2 diagnostics-12-02131-f002:**
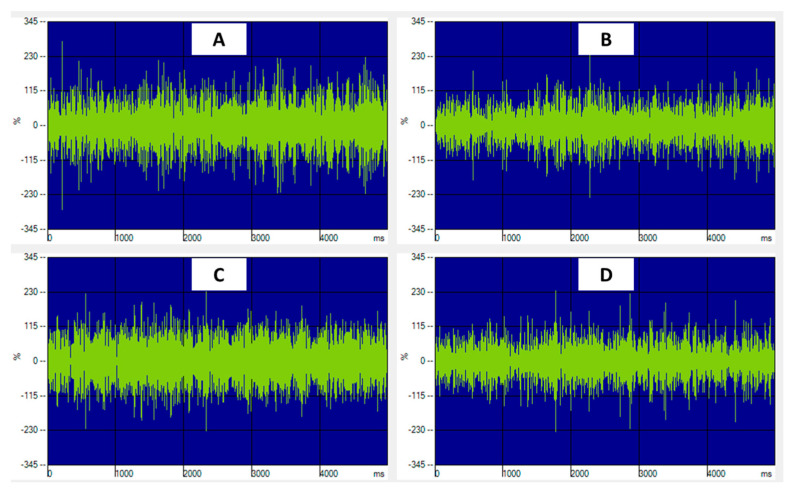
(**A**–**D**) Example of an electromyographical signal of the muscles activated during clenching. *X*-axis: time (ms), *Y*-axis: intensity of activation (μV/μV × 100). (**A**) right anterior temporalis muscle; (**B**) left anterior temporalis muscle; (**C**) right masseter muscle; (**D**) left masseter muscle.

**Table 1 diagnostics-12-02131-t001:** Mean values and standard deviation (SD) relating to age, orthodontic characteristics, and duration of the treatment in FOA and ALIGN group. Significant differences between groups at *t*-Test were set at 5% (*p* < 0.05). *p* < 0.05 *.

Index	FOA (Mean ± SD)	ALIGN (Mean ± SD)	Student *t*-Test
Age	12.49 ± 1.07	25.60 ± 13.17	*p* < 0.05 *
Sex	3 males 4 females	3 males 6 females	
Treatment duration	23.86 ± 7.69	15.33 ± 7.78	*p* < 0.05 *
SNA	80.8 ± 2.7	80.7 ± 4.8	0.96
SNB	76.9 ± 2.3	77.2 ± 4.9	0.77
ANB	3.9 ± 1.7	3.5 ± 1.3	0.35
SN^Go.Gn	34.2 ± 4.6	33.3 ± 6.8	0.79
Overjet	1.8 ± 0.8	0.8 ± 1.0	0.12
Overbite	1.8 ± 0.8	1.0 ± 0.8	0.08

**Table 2 diagnostics-12-02131-t002:** Mean and standard deviation (SD) relating to all standardized electromyographic indexes within the FOA and ALIGN groups at T1, T2, and T3 are shown: POC (%), IMPACT (%), ASIM (%), ACTIV (%), TORQUE (%) (masseter muscles MM, temporalis muscles TA). Two-way ANOVA with repeated measures on one factor was performed between groups and significance was set at 5% (*p* < 0.05).

*Index*	T1		T2		T3		*2-Way ANOVA*
	FOA (Mean ± SD)	ALIGN (Mean ± SD)	FOA (Mean ± SD)	ALIGN (Mean ± SD)	FOA (Mean ± SD)	ALIGN (Mean ± SD)	
POC TA (%)	83.4 ± 5.9	82.4 ± 8.0	83.5 ± 3.3	84.5 ± 4.1	85.3 ± 2.7	84.5 ± 3.4	ns
POC MM (%)	85.1 ± 2.1 ^1^	83.2 ± 5.7	80.4 ± 6.7 ^2^	86.0 ± 3.8	86.9 ± 2.4 ^1,2^	84.7 ± 3.2	*p* < 0.05
POC mean (%)	82.3 ± 3.8	82.8 ± 6.8	81.9 ± 4.7 ^1^	85.3 ± 2.8	86.1 ± 1.7 ^1^	84.6 ± 2.3	ns
IMPACT (%)	92.8 ± 20.4 ^1^	102.2 ± 18.0	113.5 ± 26.4 ^1^	95.9 ± 7.9	100.4 ± 22.4	106.2 ± 17.0	*p* < 0.05
ASIM (%)	4.4 ± 4.2	5.5 ± 7.1	7.0 ± 8.4	6.1 ± 5.0	4.6 ± 4.1	6.9 ± 5.3	ns
ACTIV (%)	9.5 ±5.9 ^a^	3.9 ± 2.9 ^a^	15.1 ± 11.5	6.8 ± 5.4	7.6 ± 2.8	9.6 ± 7.6	*p* < 0.05
TORQUE (%)	6.1 ± 6.6	3.3 ± 4.2	5.2 ± 2.9	4.5 ± 2.9	2.9 ± 3.1	5.1 ± 3.0	ns

STROBE Statement—checklist of items that should be included in reports of observational studies. Means with different superscript number or letters differ at post hoc tests. Superscript number: intragroup significant differences between timepoints (post hoc *t*-Test, *p* < 0.05). Superscript letter: intratimepoint significant differences between treatment groups (post hoc *t*-Test, *p* < 0.05).

## Data Availability

The data underlying this article will be shared on reasonable request to the corresponding author.

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
