# Peer review of "Neuromuscular Stability of Dental Occlusion in Patients Treated with Aligners and Fixed Orthodontic Appliance: A Preliminary Electromyographical Longitudinal Case-Control Study"

_diagnostics, 2022, doi:10.3390/diagnostics12092131_

Round 1

Reviewer 1 Report

Dear Authors, 

i think the article is very interesting. I suggest analyzing only dynamic data such as EMG values in the future. The data regarding cephalometric analysis are static, and also with these information the enrolled patients will be exposed to a biological risk.

Please add the IRB number in methods section

Author Response

Dear Reviewer,

thank you for your suggestions and comments.

The following IRB number has been followed in the materials and methods sections "The study protocol was approved by the ethics committee of the University of Milan (DG-EMG-2016)."

Reviewer 2 Report

Dear authors, 

the article is truly interesting.

Please find some suggestions:

- STROBE guidelines: please adapt the manuscript to STROBE guidelines 

-TITLE: I would suggest to change "cohort" to "case-control"

- ABSTRACT: the abbreviations (MM and TA) should be specified

- INTRODUCTION: 

page 1, lines 35-39: you should also mention the use of clear aligners in "teen" patients. For your convenience, please find some references below:

Levrini L, Tettamanti L, Macchi A, Tagliabue A, Caprioglio A. Invisalign teen for thumb-sucking management. A case report. Eur J Paediatr Dent. 2012 Jun;13(2):155-8. PMID: 22762181.

Staderini E, Patini R, Meuli S, Camodeca A, Guglielmi F, Gallenzi P. Indication of clear aligners in the early treatment of anterior crossbite: a case series. Dental Press J Orthod. 2020 Jul-Aug;25(4):33-43. doi: 10.1590/2177-6709.25.4.033-043.oar. PMID: 32965385; PMCID: PMC7510494.

Staderini E, Meuli S, Gallenzi P. Orthodontic treatment of class three malocclusion using clear aligners: A case report. J Oral Biol Craniofac Res. 2019 Oct-Dec;9(4):360-362. doi: 10.1016/j.jobcr.2019.09.004. Epub 2019 Oct 7. PMID: 31667066; PMCID: PMC6811999.

MATERIALS AND METHODS: 

page 4, lines 174 and 178: MVC is "maximum voluntary contraction" or "maximum voluntary clenching"?

page 5, line 206: please find a reference for this sentence

page 5, lines 254-256: how do you evaluate these items?

A sample size calculation should be needful.

Author Response

Dear Reviewer,

thank you for your suggestions and comments.

The following parts have been modified/added in the manuscript:

1. STROBE - you can find strobe guidelines filled in the attachment

2. The title has been changed as suggested as follows "Neuromuscular stability of dental occlusion in patients treated with aligners and fixed orthodontic appliance: A preliminary electromyographical longitudinal case-control study"

3. The abstract section has been modified as follows "Standardize surface electromyography (ssEMG) was used to evaluate the activity of masticatory muscles (masseters-MM and temporalis-TM) before the beginning of orthodontic treatment (T1), at the end (T2) and 3 months (T3) after the end of the treatment. "

4. The introduction section has been modified as follows: "They were born as an orthodontic alternative for adult patients with Angle class I malocclusion and mild to moderate crowding but to date, even more complex cases including extractions [1], [2], class II subdivision [3], openbite [4], surgical class III [5] and for interceptive treatment in children [6] have been presented."

5. at page 4, lines 174 and 178 "maximum voluntary contraction (MVC)" has been changed in "maximum voluntary clenching" has follows:

  • Masticatory muscles standardization procedure: two 10-mm thick cotton rolls were positioned on the mandibular second premolars/first molars of each patient, and a 5-s maximum voluntary clenching (MVC) was recorded to standardize TA and MM sEMG signal. The mean ssEMG potential obtained in the first acquisition was set at 100%, and all further sEMG potentials were expressed as a percentage of this value (µV/µVx100) [14].

6. page 5, line 206, the following reference has been added: Ferrario VF, Sforza C. Coordinated electromyographic activity of the human masseter and temporalis anterior muscles during mastication. Eur J Oral Sci. 1996 Oct-Dec;104(5-6):511-7. 

7. page 5, lines 254-256 the following has been added to text to clarify the functional evaluation at the end of the treatment: 

"Before concluding the therapies, all the patients of both groups were evaluated by the operators (GG and NK):

  • functional mandibular movement were evaluated clinically by asking the patients to protrude and to move laterally the lower jaw, neither deflections nor impediments were observed;
  • occlusal stability was evaluated by asking the patients to clench on articulating paper (blu articulating paper 200µ Bausch, Nashua USA) and to bite hard; nor shift or deflections were noticed, and symmetrical distribution of occlusal contacts was observed"

8. A sample size has been calculated as follows: 

"Sample Size calculation

By assuming to detect a significant statistical difference in the POC values between the 2 groups of 4.59% with a standard deviation of 5.24 as previously reported by Ferrario, a margin error of 22.8 was calculated. Considering a power of 80%, alpha 0.05, beta 0.02 a sample size of 30 was detected. Considering the nature of this study, preliminary data in a sample of 16 patients are herein presented."
